# The transmission mechanism of the housing price fluctuations on the global value chain position of manufacturing-evidence from China

Jingyun Zhang[1], Guangping Liu[2]*, Xueyuan Li[3], Han Jiang[2]

**1** College of Economics and Management, Nanjing University of Aeronautics and Astronautics, Nanjing, China, **2** School of Economics and Management, Hebei University of Technology, Tianjin, China, **3** College of Management and Economics, Tianjin University, Tianjin, China

\* lgp@hebut.edu.cn

**Data Availability Statement:** The data underlying this study belong to the National Bureau of Statistics of China. Data on housing price can be obtained from http://www.stats.gov.cn/tjsj/ndsj/

## Abstract

Since the real estate market reform in 1998, China's house prices have been rising. High housing prices have exerted negative impacts on labour mobility, the investment environment and resident consumption, all of which may affect the overall resource allocation efficiency and the improvement of the global value chain position of China's manufacturing industry. However, there is little research on the mechanism of the impact of housing prices on the status of manufacturing global value chain. Based on the matching data of China and the *OECD-TiVA* from 2005 to 2016, the parallel multiple mediator model is adopted to empirically test the transmission mechanism of housing price fluctuation on the global value chain position of China's manufacturing industry in this study using human capital level, resident consumption level, resident consumption structure, and R&D investment level as the mediating variables. This article finds that although housing price fluctuations do not have a direct impact on the global value chain position of the manufacturing industry, human capital level and resident consumption structure do have positive and negative mediating effects, respectively. Finally, we put forward some suggestions to promote the global value chain position of China's manufacturing industry, which provides useful reference for policy makers.

## Introduction

Since China joined the WTO, the manufacturing industry has maintained rapid development and its participation in the international division of labour has steadily increased. The GDPs of the Chinese manufacturing industry were 0.3469 and 2.6482 trillion yuan in 2001 and 2018, respectively, the average annual growth rate was 12.7% [1–2]. Currently, China's export products primarily include the processing of semi-finished products and small commodities such as toys and clothing. That said, it is well known that China's international competitiveness with respect to high-tech products must be improved as its manufacturing industry's

and data on global value chain position of manufacturing can be obtained from http://www.oecd.org/industry/ind/measuring-trade-in-value-added.htm. Please see the Supporting Information of this article for specific details related to data curation. The authors did not have special access privileges.

**Funding:** Financial support from the Hebei Soft Science Foundation [grant number 19456106D], the China Postdoctoral Science Foundation [grant number 2017M621047], 14th Five-Year Plan Foundation of Hebei Development and Reform Commission [grant number DRC14-21], and Hebei Social Science Foundation [grant number HB18GL034]. The funders had no role in study design, data collection and analysis, decision to publish, or preparation of the manuscript.

**Competing interests:** The authors have declared that no competing interests exist.

competitive advantage relies primarily on lower labour costs and larger export volumes. Accordingly, the position of China's manufacturing industry in the international division of labour is concentrated, for the most part, in the middle and lower links of the value chain due to the industry's low added value and low-level technology products. However, with the increase in labour costs, China's traditional labour advantage is not sustainable over the long term because the traditional model consumes such a large amount of domestic resources, a situation that constrains development sustainability. Therefore, exploring the influencing factors of global value chain and its promotion path is the key to promote the high-quality development of China's manufacturing industry and enhance its global value chain status. At present, although there are many scholars to study this problem and put forward the corresponding recommendations, but few studies from the perspective of housing price fluctuations to analyse.

After China started its market-oriented real estate reform in 1998, especially from 2005 to 2016, housing prices rose rapidly. During this period, China's rapid economic development, the acceleration of urbanization, and the annual growth of residents' income make people's demand for housing more and more. Good economic fundamentals and residents' income growing with economic growth provide a solid support for the rise of house prices, and the real estate industry has become an important pillar to promote China's economic growth. At the same time, high housing prices have exerted negative impacts on labour mobility, the investment environment and resident consumption, all of which may affect the overall resource allocation efficiency and hinder the development of manufacturing at a higher level, especially the improvement of the global value chain position of China's manufacturing industry. From 2005 to 2016, the correlation coefficient between China's housing prices and the global value chain position of the manufacturing industry was as high as 0.94 (see Fig 1). Nonetheless, whether China's housing prices have an impact on the manufacturing industry's position on the global value chain and the internal mechanism of housing price fluctuations remains uncertain.

This paper uses a parallel multiple mediator model to analyse the impact mechanism of housing price fluctuations on the global value chain position of manufacturing enterprises. Solving this problem is not only conducive to improving the housing price spillover effect and the global value chain position theory of manufacturing, but it will also provide guidance for the government to formulate policies that will promote the position of China's manufacturing

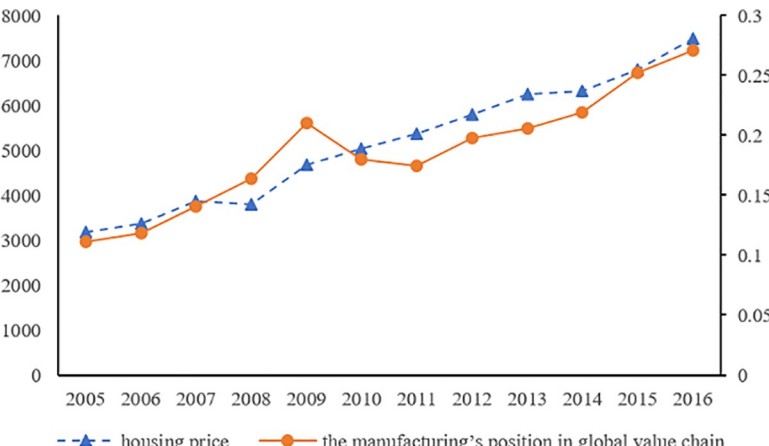

**Fig 1. Relationship between China's housing prices and the manufacturing industry's position on global value chain (note: housing price' unite is CNY/m$^2$).**

industry on the global value chain in the context of housing price fluctuations. The rest of this paper is organized as follows. Section 2 provides a brief overview of the literature on the influencing factors of global value chain and its upgrading path. Section 3 introduces the data sources, variable selection and description, and our model. Section 4 constructs the model and carries on the empirical analysis, and gives the regression results. Section 5 provides policy recommendations.

## Literature review and research hypothesis

**Literature review.** The existing studies analyse the influencing factors and improvement paths of the global value chain position from three perspectives (see Table 1).

The first perspective begins with an actual case to analyse the upgrade path of the global value chain. The global value chain links different parts of the production process to various companies around the world. Gereffin [3] analyses the clothing industry in East Asia and identifies the upgrade path of the global value chain, namely, the upgrade path from OEA (original equipment assembly), OEM (original equipment manufacturing), and ODM (original design manufacturing) to OBM (original brand manufacturing). Similarly, Humphrey et al. [4] analyse the industry of the Brazilian shoe cluster of Sinos Valley and argue that corporate relationships in the global value chain can be grouped into four types, namely, market relationships to maintain distance, networks, quasi-hierarchical structures, and hierarchical structures. There are differences in the hierarchical relationships and in control power between enterprises based on the relationships themselves. The authors further summarize the four ways to upgrade the value chain, namely, process upgrade, product upgrade, function upgrade, and chain upgrade. Burger et al. [5] conduct a specific study on the value chain upgrade path of European multinational subsidiaries and find that the breadth and scope of function upgrades are important factors in increasing the value of multinational subsidiaries.

The second perspective explores the root causes of value from factor endowments. Some scholars first analysed this issue from the theoretical level. Kaplinsky et al. [6] contend that the reason enterprises participate in different parts of the global value chain and thus create different values is because of their own unique endowments, which further generate economic rent. Lin et al. [7] argue that only by organizing production activities according to the comparative advantages of the economy can enterprises, as well as the entire country's economy, maximize an economic surplus. The above viewpoints maintain that the organization of the value chain is based on the comparative advantage of the endowment of the main factor, which lays the foundation for the subsequent research regarding the specific influencing factors of position on the value chain. Later, some scholars studied the specific factors affecting the position of the manufacturing industry on the value chain. Hu et al. [8] find that economies of scale, R&D investment, good financing conditions and foreign direct investments promote the position of the division of labour. In high-tech industries, human resource endowments have a positive impact on export prices and on the position of the division of labour. By analysing relevant theories, Li [9] empirically studies the impact of factor endowment structure, the supporting level of domestic production and the share of industry exports on value chain position and find that the first two factors are positively correlated with value chain position, while the latter is negatively correlated. Wang et al. [10] notes that the increase in labour costs in developing countries also promotes an enterprise's position on the value chain.

The third perspective considers the value chain as a whole and identifies factors that enhance value chain position. Kogut [11] notes that enterprises, theoretically, can outsource supporting activities to promote enterprise specialization, while the supporting activities are performed by specialized service industries that are separate from the original enterprise. With

**Table 1. The review of mainstream and other theoretic systems.**

| perspective | Main research issues | Main ideas and conclusions |
|---|---|---|
| Case study | analyse the upgrade path of the global value chain | Gereffin (1999) analyses the clothing industry in East Asia and identifies the upgrade path of the global value chain from equipment assembly to brand manufacturing;<br>Humphrey et al. (2002) analyse the industry of the Brazilian shoe cluster of Sinos Valley and argue that corporate relationships in the global value chain can be grouped into four types. There are four ways to upgrade the global value chain;<br>Burger et al. (2018) conduct a specific study on the value upgrade path of European multinational subsidiaries and find that the breadth and scope of function upgrades are important factors in increasing the value of multinational subsidiaries. |
| factor endowments | explores the root causes of value and the influencing factors of the global value chain position | Kaplinsky et al.(1999) contend that the reason enterprises participate in different parts of the global value chain and thus create different values is because of their own unique endowments, which further generate economic rent;<br>Lin et al.(2003) argue that only by organizing production activities according to the comparative advantages of the economy can enterprises, as well as the entire country's economy, maximize an economic surplus;<br>Hu et al. (2013) find that economies of scale, R&D investment, good financing conditions and foreign direct investments promote the position of the division of labour. In high-tech industries, human resource endowments have a positive impact on export prices and on the position of the division of labour;<br>Li (2015) empirically studies the impact of factor endowment structure, the supporting level of domestic production and the share of industry exports on value chain position;<br>Wang et al. (2015) notes that the increase in labour costs in developing countries also promotes an enterprise's position on the value chain. |
| the value chain as a whole | identifies the promotion factors of value chain position | Kogut (1985) notes that enterprises, theoretically, can outsource supporting activities to promote enterprise specialization;<br>Mohan (2016) finds that factors related to institutional level, such as rules, strategies, organizations and informal norms, have implications for value chain upgrading and the welfare of participants;<br>Ma et al. (2017) argue that the development of a productive service industry and the development of the supporting capacity of domestic production form the core competitiveness of the manufacturing industry, improving the overall operational efficiency of the value chain. |

respect to the empirical research about the impact of the institution's level and efficiency on the improvement of value chain position, Ma et al. [12] argue that the development of a productive service industry and the development of the supporting capacity of domestic production form the core competitiveness of the manufacturing industry and improve economies of scale in manufacturing, while also improving the overall operational efficiency of the value chain, which then significantly increases the global value chain position of China's

manufacturing industry. Mohan [13] finds that factors related to institutional level, such as rules, strategies, organizations and informal norms, have implications for value chain upgrading and the welfare of participants.

In summary, scholars have conducted extensive and in-depth research on improving value chain position, but they have not yet found an explanation regarding the position of the manufacturing industry on the global value chain from the perspective of housing price fluctuations. Moreover, evidence of the inherent impact mechanism of housing price fluctuations on the global value chain position of the manufacturing industry is relatively scarce. This article studies the transmission mechanism of the impact of housing price fluctuations on the global value chain position of China's manufacturing industry from four perspectives, i.e., human capital, resident consumption level, resident consumption structure and R&D investment, to enrich the theory of value chain position and provide effective policy suggestions to promote the global value chain position of China's manufacturing industry.

**Research hypothesis.** According to the theory of labour mobility, housing is the most important cost from among the numerous migration costs migrant workers in cities encounter. More specifically, fluctuations in housing prices affect the decision-making of labour migration and as a result, these fluctuations ultimately lead to changes in regional human capital. As an endogenous accumulation factor, human capital determines the level of production technology in a region, facilitates regional access to technological advantages, changes the passive situation in the global value chain production network, and is conducive to improvements in global value chain position [14]. Human capital also determines the region's technological absorptive capacity and ability to transform technological achievements, thus resulting in differences in the absorption and transformation of imported technologies in different regions and industries, which, in turn, results in differences in the region's position on the global value chain [15]. In addition, the externalities of human capital have a subtle influence on regional R&D efficiency and production efficiency [16–17], thereby affecting the region's position on the global value chain. According to the above analysis, the following hypothesis is proposed:

**H1:** Human capital plays a mediating role in the impact of housing price fluctuations on the manufacturing industry's position on the global value chain.

Housing price fluctuations have a wealth effect as well as a crowding out effect on resident consumption [18]. Specifically, for those who have housing, the rise in housing prices leads to an increase in their property income, thus promoting resident consumption, which is manifested as a wealth effect. For those who have no housing, the cost of buying or renting a house is part of their autonomous consumption. Although the rise in housing price increases the housing consumption of homeless people, it crowds out other types of consumption. This fluctuation in housing prices also affects the consumption structure of residents. When housing prices increase, consumption increases for those who have housing, and thus, their autonomous consumption is, for the most part, unaffected, while increased consumption is recognized as high-end consumption. Hence, the overall consumption structure tends to be high-end. For those without housing, the rising housing prices lead to an increase in autonomous consumption, while their high-end consumption is limited, which is not conducive to improving the consumption structure.

The changes in resident consumption level and resident consumption structure caused by housing price fluctuations further affect the global value chain position of the manufacturing industry. Because local manufacturers rely heavily on foreign demand when they enter the global market, they can achieve a certain degree of process and product upgrades. However, local manufacturers also want to integrate into the international market and achieve function and chain upgrades, but they are restricted from doing so by foreign demand. This unequal governance mode of the global value chain caused by a buyer's market hinders, to some extent,

local manufacturers from value chain upgrades. To improve this unequal value chain governance mode, it is important to stimulate consumption demand. In the case of limited foreign demand, the growth of domestic consumption demand opens the advanced market and promotes value chain upgrading. Thus, local manufacturers rely on the domestic market to establish a national value chain. To resolve the excess capacity of existing production lines and promote new products, the national value chain further absorbs the demand of surrounding regions and then re-establishes the governance mode of the network value chain to form a regional value chain, which finally integrates into the global value chain, thus achieving powerful functional and chain upgrades. According to the above discussion, two hypotheses are proposed:

**H2:** The level of resident consumption plays a mediating role in the impact of housing price fluctuations on the global value chain position of the manufacturing industry.

**H3:** The structure of resident consumption plays a mediating role in the impact of housing price fluctuations on the global value chain position of the manufacturing industry.

Housing price fluctuations affect the R&D investment of manufacturing enterprises in at least three ways. First, inflation caused by rising house prices increases costs to the manufacturing enterprises, causes manufacturing enterprises to make more conservative investment decisions, and induces them to reduce or withhold investments in R&D [19]. Second, housing price fluctuations may change the investment directions of manufacturing enterprises. As housing prices continue to rise, the bubble caused by the booming real estate industry attracts investments from other industries, thus reducing R&D investments [20]. Third, as housing prices rise, banks increase lending to the real estate industry, which improves the difficulty for manufacturing enterprises to obtain financing, thus decreasing their investments in R&D.

R&D investment as a capital factor has an important impact on the value chain upgrading of manufacturing enterprises. Specifically, the increase in R&D investment can improve the independent innovation capabilities and the transformation of technological achievements [21], which are key factors in improving the core competitiveness of enterprises [22–23]. Accordingly, it is concluded that R&D investment plays an important role in advancing the position of manufacturing enterprises on the global value chain. According to the above discussion, the following hypothesis is proposed:

**H4:** R&D investment plays a mediating role in the impact of housing price fluctuations on the global value chain position of the manufacturing industry.

## Variables and data

**Variable selection and measurement.** This article focuses on how housing price fluctuations affect the global value chain position of the manufacturing industry. In this study, the housing price is the independent variable, the global value chain position of the manufacturing enterprises is the dependent variable, and human capital, resident consumption level, resident consumption structure and R&D investment are the mediating variables. The control variables include economies of scale, foreign direct investments, factor structure and financing constraints as control variables. The definition and description of each variable are presented in Table 2.

1. Independent variable: housing price
   In this article, we use the national average sales price of commercial houses to represent the housing price. Thus,

$$national\ average\ sales\ price\ of\ commercial\ house = \frac{national\ sales\ of\ commercial\ house}{national\ saleable\ area\ of\ commercial\ house} \quad (1)$$

**Table 2. Definition and description of variables.**

| Code | Definition | Description | unit |
|---|---|---|---|
| $HP$ | Housing price | National average sales price of commercial house | CNY/m$^2$ |
| $GVC$ | Global value chain position in manufacturing | GVC position index | / |
| $HC$ | Human capital | National average years of schooling | year |
| $RCL$ | Resident consumption level | National consumer price index | / |
| $RCS$ | Resident consumption structure | The ratio of enjoyment consumption and developing consumption to total consumption | / |
| $RDI$ | R&D investment level | The ratio of R&D expenditure to GDP | / |
| $Contr_1$ | Economies of scale | The ratio of the total output value of industrial enterprises above designated size to the number of enterprises | / |
| $Contr_2$ | Foreign direct investment | Foreign direct investment in China | CNY |
| $Contr_3$ | Factor structure | The ratio of total assets of industrial enterprises above designated size to the average number of employees in all industries | / |
| $Contr_4$ | Financing constraint | The ratio of interest expense to total liabilities of industrial enterprises above designated size | / |

2. Dependent variable: global value position of manufacturing industry
We use the GVC position index to measure the global value position of the manufacturing industry. The GVC position index reflects the relative relationship between domestic indirect value added and foreign added value in the value added of exports from a certain sector of a certain country. The equation is as follow:

$$GVC_{ir} = ln\left(1 + \frac{IV_{ir}}{E_{ir}}\right) - ln\left(1 + \frac{FV_{ir}}{E_{ir}}\right) \tag{2}$$

where $IV_{ir}$ indicates the value added of the intermediate goods exported by industry $i$ of country $r$ to other countries, $FV_{ir}$ indicates foreign added value in the export of industry $i$ of country $r$, and $E_{ir}$ represents the total export value by industry $i$ of country $r$ in terms of added value. The more intermediate goods a country's sector exports to other countries, the greater the GVC position index and the higher the global value chain position of the sector.

3. Mediating variables:

a. Human capital level
There are three ways to measure human capital level, namely, income method, expenditure method and education index method. Since this article emphasizes the influence of migrants' skills on human capital and because the differences in migrants' skills are closely related to educational factors, we use the education index method. Accordingly, the education level of the population over a six years of age is divided into six categories, namely, never attended school, elementary school, junior high school, senior high school, junior college or undergraduate, postgraduate. The corresponding years of schooling are 0, 6, 9, 12, and 18.5, respectively. The calculation equation is as follows:

$$HC_{it} = 6E_{it} + 9J_{it} + 12S_{it} + 12U_{it} + 18.5P_{it} \tag{3}$$

where $HC_{it}$ represents the average year of schooling in the $t$ year of $i$ region and $E_{it}$, $J_{it}$, $S_{it}$, $U_{it}$ and $P_{it}$ represent the proportion of the population with education at the elementary, junior high, senior high, junior college and undergraduate, and postgraduate levels, respectively in the population aged six years and over in the $t$ year of $i$ region

b. Resident consumption level

The consumer price index (*CPI*) is used to measure the resident consumption level.

c. Resident consumption structure

Since this article investigates whether housing price fluctuations restrict the development of high-level demand, the classification method of Chen et al. [24] is adopted to classify resident consumption into three types, i.e., survival consumption, enjoyment consumption and development consumption. Furthermore, the proportion of enjoyment consumption and development consumption to the total consumption reflects changes in the consumption structure of residents.

d. R&D investment level

R&D investment level is expressed by the intensity of R&D investment and is measured by the ratio of R&D expenditures to the GDP, which better reflects the changing trend in R&D investments.

4. Control variables

a. Economies of scale

Economies of scale refers to the increase of benefits caused by the expansion of the scale in the production process, which affects the global value chain position of the manufacturing industry. Accordingly, we use the ratio of the total output value of manufacturing enterprises beyond the designated size to the number of enterprise units to reflect the degree of economies of scale.

b. Foreign direct investment

Foreign direct investments result, to a certain extent, in capital and technology spillover effects and have an impact on the global value chain position of manufacturing enterprises. Therefore, foreign direct investment is selected as the control variable.

c. Factor structure

The factor structure influences the comparative advantage, and thus affects the global value chain position of the manufacturing industry. The ratio of the total assets of the industrial enterprises beyond the designated size to the average number of employees compared to the total number of employees is used to represent the factor structure.

d. Financing constraint

Financing constraint determines the scale of financing for manufacturing, which affects the global value chain position of manufacturing enterprises by influencing R&D investment and innovation levels. We use the ratio of interest expenses to total liabilities of industrial enterprises beyond the designated size to reflect the financing constraint.

**Data sources:** Considering the availability and matching of the data, the time series data from 2005 to 2016 were selected as the research sample. Of these, housing price, resident consumption level, resident consumption structure, economies of scale, and foreign direct investment data are obtained from the 2006 to 2017 *China Statistical Yearbook*. The data regarding R&D investment level are obtained from the 2005 to 2016 *Statistical Bulletin of China's Science and Technology Funding*. The global value chain position of the manufacturing industry data is obtained from the *OECD-TiVA* database. The data with respect to human capital is compiled from the 2006 to 2017 *China Statistical Yearbook* and *China Population and Employment Statistics Yearbook*. The data regarding factor structure and financing constraint are obtained from the 2006 to 2017 *China Statistics Yearbook* and *China Industrial Statistics Yearbook*.

## Model construction and empirical analysis

**Model construction.** A parallel multiple mediator model is constructed. First, a direct effect model of the impact of housing price fluctuations and control variables on the global value chain position of the manufacturing industry is constructed (see Eq 4). If the coefficient $c$ is significant, there is a mediating effect, whereas if it is insignificant there is a suppressing effect [25–26]. However, regardless of the significance of the coefficient $c$, a follow-up test is required. Second, the impact model of housing price fluctuations using the four mediating variables, i.e., human capital, resident consumption level, resident consumption structure and R&D investment, is constructed (see Eqs 5, 6, 7 and 8). At this point, the independent variable, mediating variables, and control variables are entered into Eq 9. If the coefficients $a_i$ ($i = 1,2,3,4$) and $b_j$ ($j = 1,2,3,4$) are both significant, the indirect effect is significant and we proceed to step four. If at least one of the coefficients is not significant, then we continue processing the third step. Third, the bootstrap method is used to directly test whether the interaction term ($ab$) is significant. If it is significant, the indirect effect is significant, and we proceed to step four. If the indirect effect is not significant, the analysis is terminated. Fourth, the coefficient $c'$ of Eq 9 is tested. If it is found to be insignificant, no direct effect exists, indicating that there is only a mediating effect. If coefficient $c'$ of Eq 9 is significant, the direct effect is significant, and the analysis proceeds to step five. Fifth, comparing the signs of $ab$ and $c'$, if the signs are the same, there is a mediating effect. If the signs are different, there is a suppressing effect. The regression equations are as follows:

$$GVC = cHP + \sum_{i=1}^{4} \beta contr_i + \mu \qquad (4)$$

$$HC = a_1 HP + \sigma_1 \qquad (5)$$

$$RCL = a_2 HP + \sigma_2 \qquad (6)$$

$$RCS = a_3 HP + \sigma_3 \qquad (7)$$

$$RDI = a_4 HP + \sigma_4 \qquad (8)$$

$$GVC = c' HP + b_1 HC + b_2 RCL + b_3 RCS + b_4 RDI + \sum_{i=1}^{4} \beta' contr_i + \tau \qquad (9)$$

where Eq 4 is the regression model including the independent variable, dependent variable and control variables. Eqs 5 to 8 are the regression models between the independent variables and the mediating variables. Eq 9 is the regression model including the independent variable, mediating variables and dependent variables. $c$, $a_1$, $a_2$, $a_3$, $a_4$, $c'$, $b_1$, $b_2$, $b_3$, $b_4$, $\beta$ and $\beta'$ are the coefficients and $\mu$, $\sigma_1$, $\sigma_2$, $\sigma_3$, $\sigma_4$ and $\tau$ are the error terms.

**Descriptive statistical analysis.** The descriptive analysis of each variable from 2005 to 2016 is presented in Table 3. It is evident that the housing price varies greatly in different years, with a maximum value of 7476, a minimum value of 3168, and a standard deviation of 1410. With respect to the mediating variables, while there is little difference between the level of human capital and resident consumption structure, the level of resident consumption and R&D investment varies greatly in different years. The numerical values of the global value chain position of the manufacturing industry over the years is small, with the maximum and minimum values being 0.27 and 0.11. With respect to the control variables, while there is little difference in the financing condition during the study period, there is a substantial difference between the maximum and minimum values of foreign direct investment, economies of scale

**Table 3. Descriptive statistical analysis results.**

| Variable | Num. | Min. | Max. | Mean | S.E. | S.D. |
|---|---|---|---|---|---|---|
| Housing price | 12 | 3168.00 | 7476.00 | 5157.50 | 407.69 | 1.41E3 |
| Human capital | 12 | 7.83 | 9.14 | 8.60 | 0.14 | 0.47 |
| Resident consumption level | 12 | 5771.00 | 21228.00 | 1.26E4 | 1.51E3 | 5.25E3 |
| Resident consumption structure | 12 | 38.15 | 41.36 | 40.01 | 0.23 | 0.81 |
| R&D investment level | 12 | 1673.80 | 12144.00 | 6321.07 | 1.05E3 | 3.64E3 |
| Position of global value chain in manufacturing | 12 | 0.11 | 0.27 | 0.19 | 0.01 | 0.05 |
| Foreign direct investment | 12 | 638.00 | 1262.00 | 1021.92 | 64.93 | 224.912 |
| Economies of scale | 12 | 0.93 | 3.06 | 2.024 | 0.254 | 0.88 |
| Factor structure | 12 | 35.49 | 114.59 | 70.45 | 7.690 | 26.65 |
| Financing constraint | 12 | 0.01 | 0.03 | 0.02 | 0.001 | 0.003 |

and factor structure. The above analysis indicates that the Chinese housing prices, resident consumption, R&D investments, the global value chain position of the manufacturing industry, foreign direct investments, economies of scale and factor structure fluctuate greatly among the different periods.

**Empirical analysis.** The empirical results of Eq 4, which are presented in Table 4, indicate that the coefficient of the independent variable *HP* is not significant. Thus, we test the coefficients ($a_i$ and $b_j$) of Eqs 5, 6, 7, 8 and Eq 9 (see Table 4 and Table 5). The two tables suggest that only the coefficients of *HC* in Eqs 5 and 9 are significant, which indicates that the indirect effect is significant. With respect to the coefficients of the other mediating variables, only one is determined to be significant. Thus, it is necessary to use the bootstrap method to test for the existence of mediating effects and to compare the importance of the different mediating effects [27]. First, using the repeated sampling technique, 12 groups of samples with the same number of original samples are selected. The estimated value of the mediating effect $\hat{a}\hat{b}$ is then calculated based on the extracted samples. Subsequently, the above process is repeated 5000 times to obtain an estimate of 5000 mediating effects, which are then sorted from small to large, yielding sequence *B*. Finally, the 5th and 95th percentiles of sequence *B* are selected at the significance level of $p = 0.1$ to establish the 90% bootstrap confidence interval. If the bootstrap confidence interval does not contain 0, it means that there is a significant mediating effect; otherwise, the mediating effect is not significant. Specifically, the coefficient $\sum_{i=1}^{4} a_i b_i$ is tested, the significance of the overall mediating effect is assessed according to the bootstrap confidence interval, and the single mediating effect coefficient $a_i b_i$ is then tested. For data processing we use SPSS data analysis software and *Process* plug-in. The test results of the bootstrap method are displayed in Table 6.

**Table 4. Regression results of Eqs 4, 5, 6 and 7.**

| variable | Eq 5 | Eq 6 | Eq 7 | Eq 8 |
|---|---|---|---|---|
| | HC | RCL | RCS | RDI |
| HP | 0.96*** | 0.99*** | 0.55* | 0.96*** |
| $R^2$ | 0.92 | 0.98 | 0.30 | 0.91 |
| MSE | 0.08 | 0.02 | 0.77 | 0.10 |
| F−statistic | 122.95*** | 449.13*** | 4.35* | 105.46*** |

Note

*, ** and *** denote that the statistics are significant at the 10%, 5% and 1% levels, respectively

**Table 5. Regression results of Eqs 3 and 8.**

| variable | Eq 4 | Eq 9 |
|---|---|---|
| HP | 1.19 | 1.10 |
| HC | | 1.81* |
| RCL | | 3.14 |
| RCS | | -0.08 |
| RDI | | 0.45 |
| $Contr_1$ | -1.22* | -1.49** |
| $Contr_2$ | 0.07 | -0.46 |
| $Contr_3$ | 0.73 | -3.41 |
| $Contr_4$ | 0.19 | -0.32 |
| $R^2$ | 0.98 | 1.00 |
| MSE | 0.06 | 0.01 |
| $F-statistic$ | 5.00*** | 90.92** |

Note

*, ** and *** denote that the statistics are significant at the 10%, 5% and 1% levels, respectively Table 5.

Table 6 indicates that the bootstrap confidence interval of the total mediating effect ($\sum_{i=1}^{4} a_i b_i$) contains 0, meaning that the total mediating effect is not significant. Moreover, based on the perspective of each path, i.e., the housing price fluctuation-human capital level-global value chain position of the manufacturing industry path and the housing price fluctuation-resident consumption structure-global value chain position of the manufacturing industry path, the bootstrap confidence intervals do not contain 0, and thus, the mediating effects are significant. More specifically, the impact coefficients of housing price fluctuation on human capital level and resident consumption structure are 0.96 and 0.55, respectively, and the impact coefficients of human capital level and resident consumption structure on the global value chain position of the manufacturing industry are 1.81 and -0.08, respectively. The mediating effects of housing price fluctuation on the global value chain position of the manufacturing industry according to the two paths of human capital level and resident consumption structure are 1.71 and -0.04, respectively. Furthermore, the bootstrap confidence interval of housing price fluctuation-resident consumption level-global value chain position of the manufacturing industry and housing price fluctuation-R&D investment-global value chain position in the manufacturing industry contains 0, indicating that the mediating effect is not significant. The HP coefficient of Eq 8 in Table 4 further reveals that because the coefficient is not significant, a direct effect of the significant mediating effect does not exist. Accordingly, the above analysis supports hypothesis 1 and hypothesis 3, which pass the test, and disproves hypothesis 2 and hypothesis 4, which fail the test.

**Table 6. The significant test results of the mediating effect using the bootstrap method.**

| path | boot variance | indirect effect of standardization | 90% confidence interval | |
|---|---|---|---|---|
| | | | LLCI | ULCI |
| Human capital level as mediating variable | 0.99 | 0.96×1.81 = 1.71 | 0.65 | 2.32 |
| Resident consumption level as mediating variable | 9.90 | 0.99×3.14 = 3.10 | -32.99 | 6.65 |
| Resident consumer structure as mediating variable | 0.12 | 0.55×(-0.08) = -0.04 | -0.21 | -0.01 |
| R&D investment level as mediating variable | 0.56 | 0.96×0.45 = 0.43 | -0.75 | 1.80 |
| Total mediating effect | 10.37 | 1.71+3.10−0.04+0.43 = 5.23 | -3.64 | 10.10 |

Specifically, the two influencing coefficients of the mediating variable of human capital level are positive, indicating that an increase in housing price promotes the improvement of human capital and that the improvement of human capital promotes the global value chain position of the manufacturing industry. While the total influence coefficient of the mediating variable on resident consumption structure is negative, the influence coefficient of housing price fluctuation on resident consumption structure is positive, indicating that although an increase in housing price can increase the consumption structure of residents, it does not have a significant influence on the global value chain position of the manufacturing industry, which may be related to the current international division of labour. In some mid-to-high-end markets, China is still in the middle and lower levels of the global value chain, which indicates that the increased high-end demand is derived primarily from foreign added value. However, the mediating effects of resident consumption level and R&D investment are not significant, which is inconsistent with the research hypothesis. With respect to the level of resident consumption, because the consumption of residents involves multiple consumptions, China's household consumption is dominated by low-end rather than high-end consumption. That being the case, low- and middle-end consumption has less impact on the global value chain position of the manufacturing industry. The insignificance of R&D investment may be related to the direction of R&D investments in the manufacturing enterprises. Specifically, low-end R&D investment has a limited effect on the global value chain position of the manufacturing industry. In addition, as China's R&D investment efficiency is relatively low, there is a gap between increases in R&D investment and improvement in the global value chain position of the manufacturing industry.

## Conclusions and suggestions

This article uses human capital level, resident consumption level, resident consumption structure and R&D investment as mediating variables to analyse the impact of housing price fluctuations on the position of China's manufacturing industry on global value chain. The results indicate that only human capital level and resident consumption structure mediate the effects of housing price fluctuations on the industry's position on the global value chain and that these two mediating variables play positive and negative mediating roles, respectively. Moreover, the mediating effects of resident consumption level and R&D investment level are concluded to not be significant, which can be explained by the fact that Chinese resident consumption is mainly concentrated on low- and middle-end products where R&D investment efficiency is not high.

According to the research conclusions presented herein, we put forward the following suggestions to improve China's global value chain position in the manufacturing industry. First, it is recommended that the human capital level be improved while the housing prices be controlled. Although housing price increases can improve the level of human capital, they can also have many negative effects that are not conducive to the healthy development of the national economy. Highly educated people in China, especially those who have obtained doctoral degrees, work primarily in research institutions and universities, and thus, their wages are relatively low. Accordingly, China's government should encourage people to pursue high academic degrees by increasing the economic benefits for doing so. At the same time, China's manufacturing enterprises should pay attention to the innovation of human resource management methods to improve the use efficiency of human capital, increase investment in education and strengthen personnel skills training, and provide better development prospects or better job opportunities for the imported talents. Second, China should actively guide consumers to buy domestic high-end products. In other words, China's manufacturing enterprises

must strengthen their own brand building to gain the support of domestic consumers. They can implement innovative marketing strategies and increase brand awareness through cooperation with international brands. In addition, with respect to government procurement behaviours, it is essential that the government support local brands and prioritize them, providing they meet the various requirements. Third, it is recommended that China's government strengthen its high-end R&D and product promotion by constructing R&D institutions, promoting the connection among enterprises, universities and other scientific research institutions, and actualizing the combination of production, education and research. In addition, enterprises should strengthen their research capabilities, especially the development of core technologies, and the government should strengthen the cultivation of independent innovation capabilities of enterprises, support enterprises in their development of independent intellectual property rights and market-competitive process products, and increase the protection of intellectual property rights to ensure that enterprises' R&D investments result in enhanced benefits. Moreover, the R&D of enterprises should be closely combined with the market demand, so as to produce marketable products, so that the follow-up product promotion and marketing will be smooth.

## Supporting information

**S1 File. Explanation of Data.**
(DOCX)

## Acknowledgments

We thank the anonymous referees for their helpful comments and suggestions.

## Author Contributions

**Conceptualization:** Guangping Liu.

**Data curation:** Han Jiang.

**Funding acquisition:** Guangping Liu.

**Supervision:** Guangping Liu.

**Writing – original draft:** Xueyuan Li.

**Writing – review & editing:** Jingyun Zhang.

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
