## [Decision Letter · Decision Letter 0]

20 Dec 2019

PONE-D-19-32876

The Transmission Mechanism of the Housing Price Fluctuations on the Global Value Chain Position of Manufacturing

PLOS ONE

Dear Dr. Liu,

Thank you for submitting your manuscript to PLOS ONE. After careful consideration, we feel that it has merit but does not fully meet PLOS ONE’s publication criteria as it currently stands. Therefore, we invite you to submit a revised version of the manuscript that addresses the points raised during the review process.

Though a reviewer rejected PONE-D-19-32876, the reviewer provided some valuable and constructive comments. Considering reviewers’ useful comments and the interesting topic of the manuscript, I would like to give you a chance to revise your manuscript. The revised manuscript will undergo the next round of review by the same reviewers.

We would appreciate receiving your revised manuscript by Feb 03 2020 11:59PM. To enhance the reproducibility of your results, we recommend that if applicable you deposit your laboratory protocols in protocols.io, where a protocol can be assigned its own identifier (DOI) such that it can be cited independently in the future. For instructions see: http://journals.plos.org/plosone/s/submission-guidelines#loc-laboratory-protocols

We look forward to receiving your revised manuscript.

Kind regards,

Baogui Xin, Ph.D.

Academic Editor

PLOS ONE

Journal Requirements:

3. Please ensure that the title of your submission is on the title page within your main document.  Please also amend either the title on the online submission form (via Edit Submission) or the title in the manuscript so that they are identical.

Reviewers' comments:

Reviewer's Responses to Questions

**Comments to the Author**

1. Is the manuscript technically sound, and do the data support the conclusions?

Reviewer #1: Yes

Reviewer #2: Yes

Reviewer #3: Yes

2. Has the statistical analysis been performed appropriately and rigorously? 

Reviewer #1: Yes

Reviewer #2: Yes

Reviewer #3: Yes

3. Have the authors made all data underlying the findings in their manuscript fully available?

Reviewer #1: Yes

Reviewer #2: Yes

Reviewer #3: Yes

4. Is the manuscript presented in an intelligible fashion and written in standard English?

Reviewer #1: Yes

Reviewer #2: Yes

Reviewer #3: Yes

5. Review Comments to the Author

Reviewer #1: Your paper analysis the effect of HP on GVC and get good conclusions and put forward excellent suggestions, which help us to understand the relationship bewteen HP and GVC. But GVC may affect the HP. So you may consider the endogeneity of the model if possible.

Reviewer #2: 1. Fig.1, table2 should be marked with units.

2. The characters of "house price" in China should be explained. The story in China is quite special, housing price raised rapidly since housing reform in 1998, especially during 2005-2016. Statistics show that China's household real estate accounts for 69% of total assets. The author is suggested to explain these backgrounds for the readers to better understand the situation.

3. The suggestions in the conclusion part can be further improved in terms of operability and innovation. For example, what other ways can we improve the level of human capital in addition to improving treatment? Can we consider career development and so on. Suggest the author to put forward more targeted and operational suggestions.

Reviewer #3: Abstract should be rewritten. You can start with one or two background sentences. Next you need to clarify the problem, and state the research gaps. Then explain how you are going to bridge this gap. Abstract should be continues with presenting the most important findings in your research. One or two concluding sentences about the value of your work and the area of application is highly recommended.

Introduction part should be restructured. It can be done in different ways. Because the authors targeted practical implications, first paragraph should provide a background to the problem (or introduce a research question), and the following few paragraphs should brief the MOST relevant studies in that specific field with a critical view. The final paragraph must then clarify the contribution(s) of the study. Quality of introduction part is of certain cachet to attract readers to read the entire paper. Given the importance of global value chain topics, this view point can be included in the first two paragraphs. Including additional references can add value.

Literature Review: The authors are highly recommended to take some time and prepare a well-structured table for the review of mainstream and other theoretic systems. Please pay attention to the fields you select for this abstract review. Some tables will help reviewer to have a glimpse of your work; it adds value to your paper and support the research gaps you stated in the introduction part. A concluding paragraph in this section can clarify the research contributions based on the identified gaps in the literature.

Before you implement your empirical research, you should give some classical economics theories which support your conclusion. In fact, using real data, we can verify any correlations in the real word without any true theories base though these relationships are whimsical.

6. PLOS authors have the option to publish the peer review history of their article (what does this mean?). If published, this will include your full peer review and any attached files.

Reviewer #1: No

Reviewer #2: No

Reviewer #3: No

---

## [Author Response · Author response to Decision Letter 0]

16 Jan 2020

Dear reviewer and editor,

we revised this manuscript according to your suggestions, see the attachment for details.

best regards,

Guangping

---

## [Decision Letter · Decision Letter 1]

21 Jan 2020

The Transmission Mechanism of the Housing Price Fluctuations on the Global Value Chain Position of Manufacturing-Evidence from China

PONE-D-19-32876R1

Dear Dr. Liu,

We are pleased to inform you that your manuscript has been judged scientifically suitable for publication and will be formally accepted for publication once it complies with all outstanding technical requirements.

With kind regards,

Baogui Xin, Ph.D.

Academic Editor

PLOS ONE

Additional Editor Comments (optional):

Reviewers' comments:

Reviewer's Responses to Questions

**Comments to the Author**

1. If the authors have adequately addressed your comments raised in a previous round of review and you feel that this manuscript is now acceptable for publication, you may indicate that here to bypass the “Comments to the Author” section, enter your conflict of interest statement in the “Confidential to Editor” section, and submit your "Accept" recommendation.

Reviewer #3: All comments have been addressed

2. Is the manuscript technically sound, and do the data support the conclusions?

Reviewer #3: Yes

3. Has the statistical analysis been performed appropriately and rigorously? 

Reviewer #3: Yes

4. Have the authors made all data underlying the findings in their manuscript fully available?

Reviewer #3: No

5. Is the manuscript presented in an intelligible fashion and written in standard English?

Reviewer #3: Yes

6. Review Comments to the Author

Reviewer #3: The authors have modified the whole paper and propose reasonable explanations according to our comments. I suggest that this paper can be published in PLOS ONE. Thank you for your contribution!

7. PLOS authors have the option to publish the peer review history of their article (what does this mean?). If published, this will include your full peer review and any attached files.

Reviewer #3: No

---

## [Editor Report · Acceptance letter]

18 Feb 2020

PONE-D-19-32876R1 

The Transmission Mechanism of the Housing Price Fluctuations on the Global Value Chain Position of Manufacturing-Evidence from China 

Dear Dr. Liu:

I am pleased to inform you that your manuscript has been deemed suitable for publication in PLOS ONE. Congratulations! Your manuscript is now with our production department. 

With kind regards,

on behalf of

Prof. Baogui Xin 

Academic Editor

PLOS ONE